# Understanding COVID-19 vaccine hesitancy in Meghalaya, India: Multiple correspondence and agglomerative hierarchical cluster analyses

**Sooyoung Kim**[1☯], **Rajiv Sarkar**[2☯], **Sampath Kumar**[3], **Melissa Glenda Lewis**[2], **Yesim Tozan**[4]*, **Sandra Albert**[2]*

1 Department of Public Health Policy and Management, School of Global Public Health, New York University, New York, New York, United States of America, 2 Indian Institute of Public Health Shillong, Pasteur Hill, Shillong, Meghalaya, India, 3 Health and Family Welfare Department, Government of Meghalaya, Additional Secretariat, Shillong, Meghalaya, India, 4 Department of Global and Environmental Health, School of Global Public Health, New York University, New York, New York, United States of America

☯ These authors contributed equally to this work.
* tozan@nyu.edu (YT); Sandra.albert@iiphs.ac.in (SA)

**Data Availability Statement:** The data that support the findings of this study are available on request from the Indian Institute of Public Health Shillong.

## Abstract

Meghalaya, a state in the northeastern region of India, had a markedly low vaccine uptake compared to the other states in the country when COVID-19 vaccines were being rolled out in 2021. This study aimed to characterize the distinct vaccine-hesitant subpopulations in healthcare and community settings in Meghalaya state in the early days of the vaccination program. We used data from a cross-sectional survey that was administered to 200 health-care workers (HCWs) and 200 community members, who were *a priori* identified as 'vaccine-eligible' and 'vaccine-hesitant,' in Shillong city, Meghalaya, in May 2021. The questionnaire collected information on participants' sociodemographic characteristics, COVID-19 history, and presence of medical comorbidities. Participants were also asked to provide a dichotomous answer to a set of 19 questions, probing the reasons for their hesitancy towards COVID-19 vaccines. A multiple correspondence analysis, followed by an agglomerative hierarchical cluster analysis, was performed to identify the distinct clusters of vaccine-hesitant participants. We identified seven clusters: indecisive HCWs (n = 71), HCWs skeptical of COVID-19 and COVID-19 vaccines (n = 128), highly educated male tribal/clan leaders concerned about infertility and future pregnancies (n = 14), less educated adults influenced by leaders and family (n = 47), older adults worried about vaccine safety (n = 76), middle-aged adults without young children (n = 56), and highly educated ethnic/religious minorities with misinformation (n = 8). Across all the clusters, perceived logistical challenges associated with receiving the vaccine was identified as a common factor contributing to vaccine hesitancy. Our study findings provide valuable insights for local and state health authorities to effectively target distinct subgroups of vaccine-hesitant populations with tailored health messaging, and also call for a comprehensive approach to address the common drivers of vaccine hesitancy in communities with low vaccination rates.

Contact information: Indian Institute of Public Health Shillong, Lawmali, Pasteur Hill, Shillong, Meghalaya, India - 793001 Tel: 0364 2592014 Email: iiphs.iec.ner@gmail.com.

**Funding:** The authors received no specific funding for this work.

**Competing interests:** We have read the journal's policy and the authors of this manuscript have the following competing interests: 1) Dr. Rajiv Sarkar serves as an Academic Editor to PLOS Global Public Health. 2) The rest of the authors declared no competing interests.

## Introduction

Since the World Health Organization (WHO) declared COVID-19 as a pandemic in March 2020, it has affected various aspects of people's lives across the globe [1]. In India, a country with significant within-country disparities in health and socioeconomic conditions, the impact of COVID-19 pandemic was the most pronounced during its second wave, which started around late March 2021. The introduction of the Delta variant marked the country as one of the global epicenters [2,3]. Just before this surge in cases, the national COVID-19 vaccination roll-out began in January 2021 with healthcare and frontline workers given priority. By April 1, 2021, the priority gradually expanded to include any residents of age 45 years and older, while others could still receive the vaccine with an out-of-pocket co-pay [4]. From May 1, 2021, eligibility was expanded to all adults aged 18 years and above [5]. The COVID-19 vaccination program in India has been decentralized, with each state government being responsible for vaccine distribution [6]. During this period, there were significant regional disparities in both COVID-19 infection and vaccination rates. In general, COVID-19 infections were highly clustered in more urbanized and prosperous states [7–9], and so was the distribution of COVID-19 vaccines [10]. These within-country disparities were due to a myriad of factors, including population density, healthcare infrastructure, socioeconomic status, mobility, and supply chain issues, all of which collectively shaped the differential impacts of the pandemic across different regions [7–10]. On the other hand, several studies highlighted the disparities in the quality of COVID-19 data reporting, which hindered an accurate estimation of the disease burden of COVID-19 and other associated issues, including an adequate understanding of the extent of vaccine hesitancy and refusal, in less prosperous states [11,12]. The national vaccination registration page, which was available only in English, and the COVID-19 vaccination co-pay at private health facilities, which could be up to $20 per dose, also exacerbated the disparities across the country [13].

Meghalaya, a state in the northeastern region of India, has a population of around 3.44 million people and home to a large number of tribal communities, whose ethnicities are Khasi, Jaintia, and Garo tribes [14,15]. As of 2021, Meghalaya was the fifth poorest state in India with a predominantly agrarian economy [16]. Even before the COVID-19 pandemic, Meghalaya had low immunization rates for routine childhood vaccines [17]. This had been identified as an issue by the state, and Meghalaya implemented a dedicated intervention based on the principles of a state capability approach in 2020 to mobilize all the stakeholders from the state's leadership to frontline healthcare workers to communities, which led to rapid improvements in vaccination coverage [18]. Nonetheless, in the beginning of the COVID-19 vaccine roll-out in 2021, Meghalaya was one of the lowest-performing states in India with its COVID-19 vaccine coverage ranking 26[th] out of the 28 states and nine union territories [10].

In early May 2021, at a COVID-19 expert committee meeting of the state government of Meghalaya, anecdotes of high vaccine hesitancy among workers at various tertiary care hospitals in the capital city of Shillong, Meghalaya, were brought up and discussed. The WHO Strategic Advisory Group of Experts on Immunization (SAGE) Working Group defines vaccine hesitancy as "delay in acceptance or refusal of vaccination despite availability of vaccination services." [19] At the time, several studies documented high rates of COVID-19 vaccine hesitancy among healthcare workers in different parts of the world [20–24] This was particularly concerning given the crucial role of healthcare workers in promoting vaccine acceptance among vaccine-hesitant communities [25]. The evidence on the nature and extent of COVID-19 vaccine hesitancy among healthcare workers in India is limited but has been growing [26]. However, no study has focused on north-east India, including Meghalaya state, where routine vaccination rates in the general population are markedly low. Furthermore, the characteristics

of vaccine-hesitant groups in this particular setting have not been described, which are known to be vaccine- and context-specific and vary across populations and over time [19].

Against this background, our study aimed to identify the distinct subgroups of vaccine-hesitant populations in the healthcare and community settings in Meghalaya state during the early days of the COVID-19 vaccination program. The overall goal of this study was to contribute to the scant evidence that can guide public health authorities in developing appropriate interventions to address the issue of vaccine hesitancy in communities with low vaccination coverage. The issue of vaccine hesitancy has become more evident during a global pandemic and in the face of new and novel vaccines to address it. Existing studies have shown that vaccine hesitancy is associated with individuals' socioeconomic status, which can be reflected in individuals' health-related knowledge, attitudes, and beliefs in general [27,28]. Furthermore, individuals' perceived barriers and negative attitudes toward vaccination may exert unique effects on their acceptance of vaccines [29]. Under this notion, we aimed to examine the factors that might have affected COVID-19 vaccine acceptance in a setting with low overall vaccination rates, with the primary aim of identifying the distinct clusters of individuals who can be potentially targeted with tailored communication strategies. Our findings also have implications for designing targeted strategies to improve vaccination compliance among healthcare workers.

## Methods

### Ethics statement

The protocol for this study was approved by the Indian Institute of Public Health Shillong Institutional Ethics Committee (IIPHS IEC) in April 2021 (Reference no: IIPHS-IEC/2021-22/01). Written consent was obtained from health workers who participated in face-to-face interviews, and verbal consent was sought from community members who were interviewed via phone due to the COVID-19 lockdown. Additional information regarding the ethical, cultural, and scientific considerations specific to inclusivity in global research is included in the (S1 Checklist).

### Study sample and data collection

This mixed-method study, comprising of a cross-sectional survey and qualitative key-informant interviews, was conducted in May 2021 to investigate the drivers of COVID-19 vaccine hesitancy among healthcare workers and members of the wider community in Shillong city, Meghalaya, India. The cross-sectional survey was conducted among people who were *a priori* identified as potentially 'vaccine-hesitant' [20,30,31]. Based on the WHO SAGE definition of vaccine hesitancy [19], for both groups, a vaccine-hesitant individual was defined as "an individual who did not receive COVID-19 vaccination by April 15, 2021 despite having had an opportunity to receive it free of charge." By this date, COVID-19 vaccines were available to healthcare workers and other frontline workers, as well as to anyone aged 45 years and above at no cost, while the rest of the population had to co-pay to receive the vaccine. For this reason, we limited our community sample to those aged 45 years and above. The healthcare workers were recruited from four tertiary care hospitals, including the two largest state government-run hospitals and two private multispecialty hospitals in Shillong city. The second group included vaccine-hesitant members of the wider community, residing in hotspots with low COVID-19 vaccination coverage in East Kasi Hill district in Shillong city. A stratified random sampling was conducted among eligible healthcare workers using the occupation categories within the health sector (i.e., doctor, nurse, paramedical staff, frontline worker, and support staff). For community members, a list of localities in Shillong agglomeration with their

respective estimated population size and COVID-19 vaccine coverage rate was obtained from the local health authorities. Using this list, a total of 10 clusters with a reported vaccination coverage rate of less than 40% were selected using a probability-proportion-to-size (PPS) approach. We then recruited participants using a random sampling approach from the list of eligible individuals provided by the Accredited Social Health Activists (ASHAs) in each cluster. ASHAs are government-employed community based frontline health workers who form a key component of the National Health Mission (NHM) in India [32]. Informed consent was obtained from all participants prior to the survey administration and those who did not express their consent were excluded from the survey. Written consent was obtained from health workers who participated in face-to-face interviews, and verbal consent was sought from community members who were interviewed via phone due to the COVID-19 lockdown. The recruitment of participants took place between May 1st, 2021 and May 22th, 2021.

The questionnaire was administered to healthcare workers face-to-face in English by trained bilingual research staff. For community members, telephonic interviews were conducted due to travel restrictions imposed during the 'lockdown' related to the ongoing pandemic. The telephone numbers of all residents in the selected localities were obtained from the ASHAs. Potential eligible participants were randomly selected and contacted for the survey. After obtaining informed consent and ascertaining their vaccination status, interviews were conducted in the local Khasi language by trained volunteers. The questionnaire collected information on participants' sociodemographic characteristics, including age, gender, ethnicity, religion, occupation, educational attainment, and place of residence, and COVID-19 history, and presence of medical comorbidities. Participants were also asked to provide a dichotomous (yes or no) answer to a set of 19 questions (Table A in S1 Text), probing perceived barriers and negative attitudes toward COVID-19 vaccination which may influence COVID-19 vaccine acceptance among vaccine hesitant populations. Unstructured responses to describe their reasons for vaccine hesitancy were also recorded.

## Statistical analysis

We first performed a descriptive analysis to characterize the participants' demographic and socioeconomic characteristics. Using a R package *wordcloud* [33], we performed an exploratory textual analysis to investigate the stated reasons for vaccine hesitancy captured in the unstructured responses. We then performed a multiple correspondence analysis (MCA) followed by an agglomerative hierarchical cluster analysis (AHCA) to identify the unique clusters of vaccine-hesitant survey participants. We employed these explanatory analytical techniques because these techniques are well suited to analyze the pattern of relationships in a data set of multiple categorical variables. When combined, these methods can effectively capture the complex patterns in a dataset through dimensionality reduction and find clusters in a data set. Specifically, the MCA can be described as a type of principal component analysis (PCA) of categorical data [34]. In this analogy, a dataset of $n$ observations and $m$ variables is considered as a $n$ x $m$ matrix in a high-dimensional Euclidean space, and the MCA aims to reduce the dimensions of the space while capturing the most variance in the data [34]. To do so, the technique combines correlated categorical variables in the data set into continuous variables. To run the MCA, we first recoded the continuous age variable as a categorical age group variable using a 5-year step increment. In the MCA, we included participants' sociodemographic characteristics (i.e., age, gender, ethnicity, religion, occupation, and educational attainment), COVID-19 history, whether the participant had any medical comorbidity, whether the participant had any child, whether the participant's child received essential childhood immunizations and participants' responses to the 19 questions, probing their perceived barriers and negative attitudes for COVID-19 vaccination.

Subsequently, the continuous variables that resulted from the MCA were grouped using hierarchical clustering analysis, which employs a step-wide approach to classify individual observations into clusters based on the measure of similarity or distance [35,36]. There are two types of hierarchical clustering: agglomerative (bottom-up) clustering and divisive (top-down) clustering. Agglomerative clustering can be further divided into subtypes based on the clustering algorithm. In this analysis, we performed the clustering using five different clustering methods, namely arithmetic average-based agglomerative clustering, single linkage agglomerative clustering, complete linkage agglomerative clustering, Ward's agglomerative clustering, and divisive clustering. We then used the clustering coefficients from each clustering output as a criterion to determine the best clustering method to interpret the outputs. The clustering coefficient, which ranges between 0–1, describes the strength of the generated clustering structure, and values closer to 1 indicate a stronger clustering structure. Lastly, we used the sum of within-cluster inertia, calculated as a clustering output, to determine the optimal number of clusters. Specifically, the partition was determined at the point with the highest relative loss of inertia [37]. All analyses were performed using R packages *FactoMineR*, *cluster*, and *factoextra* on R version 4.2.2. [38–40] Study data is available as a supporting information (S1 Data) with PLOS Human Participants Research Checklist.

## Results

### Sample description

Table 1 presents the sociodemographic characteristics of the survey participants. Of the 400 participants, 200 were healthcare workers, and 200 were community members. The survey participants were predominantly female; 70.0% of community members and 80.0% of healthcare workers. The majority of participants identified themselves as Khasi (89.0%) and Christians (90.5%). A little over half of healthcare workers (51.0%) had a college degree or higher, and 74.0% of community members had an educational attainment below college level, with more than one third (37.5%) having completed up to grade 7, which is equivalent to less than a high school diploma. The participants from the wider community were primarily concentrated in two blocks in East Khasi Hills district, with 41.5% residing in Mawlai block and 37.5% in Mylliem block. Among healthcare workers, the majority identified themselves as nurses (40.5%), followed by frontline workers (14.0%), paramedic staff (16.5%), and support staff (24.5%).

Among healthcare workers, breastfeeding was the most common stated reason for COVID-19 vaccine hesitancy (Fig 1A). Other frequently reported reasons included fear of side effects, lack of confidence in the vaccine, a perceived lack of need for the vaccine, and a willingness to wait to be vaccinated. Among community members, the pattern of the most frequently stated keywords was less clear (Fig 1B). However, concerns about side effects and a lack of universal coverage for the vaccine were frequently mentioned by community members.

### Multiple correspondence analysis

In summary, a total of 105 variable-category combination was reduced to 73 dimensions, explaining 100% of the variance. The top 15 dimensions explained about 51% of the total variance, while the top 5 accounted for 27% of the total variance. The top 15 dimensions and their eigenvalues and the percentage and cumulative percentage of the variance explained were summarized in the supplementary material (Table B in S1 Text). Fig 2 illustrates the coordinate distribution of each variable-category combination on the first two dimensions of the MCA. Results using different dimensions were generated and presented in the supplementary material (Fig A~I in S1 Text).

**Table 1. Sociodemographic characteristics of the survey participants (N = 400).**

| | Community members (N = 200) | Healthcare workers (N = 200) | Total (N = 400) |
|---|---|---|---|
| **Age** | | | |
| Mean (SD) | 56.3 (8.74) | 34.3 (8.69) | 45.3 (14.0) |
| **Sex** | | | |
| Male | 79 (39.5%) | 41 (20.5%) | 120 (30.0%) |
| Female | 121 (60.5%) | 159 (79.5%) | 280 (70.0%) |
| **Ethnicity** | | | |
| Jaintia | 13 (6.5%) | 16 (8.0%) | 29 (7.3%) |
| Khasi | 177 (88.5%) | 179 (89.5%) | 356 (89.0%) |
| Others | 10 (5.0%) | 5 (2.5%) | 15 (3.8%) |
| **Religion** | | | |
| Christian | 175 (87.5%) | 187 (93.5%) | 362 (90.5%) |
| Niam Khasi | 15 (7.5%) | 12 (6.0%) | 27 (6.8%) |
| Others | 10 (5.0%) | 1 (0.5%) | 11 (2.8%) |
| **Occupation** | | | |
| Farmer/ Agricultural worker | 9 (4.5%) | 0 (0%) | 9 (2.3%) |
| Government Employee* | 31 (15.5%) | 0 (0%) | 31 (7.8%) |
| Headmen | 14 (7.0%) | 0 (0%) | 14 (3.5%) |
| Homemaker | 45 (22.5%) | 0 (0%) | 45 (11.3%) |
| Others | 69 (34.5%) | 0 (0%) | 69 (17.3%) |
| Private Business | 32 (16.0%) | 0 (0%) | 32 (8.0%) |
| Doctor | 0 (0%) | 9 (4.5%) | 9 (2.3%) |
| Frontline health worker | 0 (0%) | 28 (14.0%) | 28 (7.0%) |
| Nurse | 0 (0%) | 81 (40.5%) | 81 (20.3%) |
| Paramedical Staff | 0 (0%) | 33 (16.5%) | 33 (8.3%) |
| Support Staff | 0 (0%) | 49 (24.5%) | 49 (12.3%) |
| **Education** | | | |
| 0 to primary | 47 (23.5%) | 0 (0%) | 47 (11.8%) |
| Up to class 7 | 28 (14.0%) | 16 (8.0%) | 44 (11.0%) |
| Up to class 10 | 41 (20.5%) | 0 (0%) | 41 (10.3%) |
| Up to class 12 | 32 (16.0%) | 65 (32.5%) | 97 (24.3%) |
| College degree/Diploma | 52 (26.0%) | 102 (51.0%) | 154 (38.5%) |
| Postgraduate | 0 (0%) | 17 (8.5%) | 17 (4.3%) |
| **Received COVID-19 vaccine after April 15** | | | |
| No | 118 (59.0%) | 0 (0%) | 118 (29.5%) |
| Yes | 68 (34.0%) | 4 (2.0%) | 72 (18.0%) |
| Response not provided | 14 (7.0%) | 196 (98.0%) | 210 (52.5%) |
| **Currently have children Aged 0–5 years** | | | |
| No | 88 (44.0%) | 88 (44.0%) | 176 (44.0%) |
| Yes | 112 (56.0%) | 112 (56.0%) | 224 (56.0%) |
| **Children received essential immunization** | | | |
| No | 30 (15.0%) | 31 (15.5%) | 61 (15.3%) |
| Yes | 82 (41.0%) | 81 (40.5%) | 163 (40.8%) |
| Not applicable | 88 (44.0%) | 88 (44.0%) | 176 (44.0%) |
| **Block of residence** | | | |
| Mawlai | 83 (41.5%) | 0 (0%) | 83 (20.8%) |
| Mawpat | 23 (11.5%) | 0 (0%) | 23 (5.8%) |

*(Continued)*

**Table 1.** (Continued)

| | Community members (N = 200) | Healthcare workers (N = 200) | Total (N = 400) |
|---|---|---|---|
| Mawryngkneng | 19 (9.5%) | 0 (0%) | 19 (4.8%) |
| Mylliem | 75 (37.5%) | 0 (0%) | 75 (18.8%) |
| Not asked* | 0 (0%) | 200 (100%) | 200 (50.0%) |

* This does not include healthcare workers (HCW) working in the Government sector. HCWs are classified based on their occupational category within the health sector.

## Agglomerative hierarchical clustering

Based on the clustering coefficient and the inertia-based selection criterion, we chose the agglomerative hierarchical clustering algorithm based on Ward's method to generate 7 unique clusters of vaccine-hesitant participants (Fig 3).

Based on participants' sociodemographic characteristics (Table C in S1 Text) and reasons for vaccine hesitancy toward COVID-19 vaccines (Fig 4), each cluster was characterized as follows:

- Cluster 1 (n = 71): Indecisive healthcare workers

- Cluster 2 (n = 128): Healthcare workers who do not believe in COVID-19 and COVID-19 vaccine

- Cluster 3 (n = 14): Highly educated male tribal/clan head with concerns on infertility and/or adverse effect on future pregnancy

- Cluster 4 (n = 47): Adults with low educational attainment who were highly influenced by influential leaders and family members

- Cluster 5 (n = 76): Senior adults with worries about COVID-19 vaccine safety

- Cluster 6 (n = 56): Middle-aged adults with no children aged 0–5 years

- Cluster 7 (n = 8): Highly educated adults who identified themselves as ethnic and religious minorities with high level of misinformation

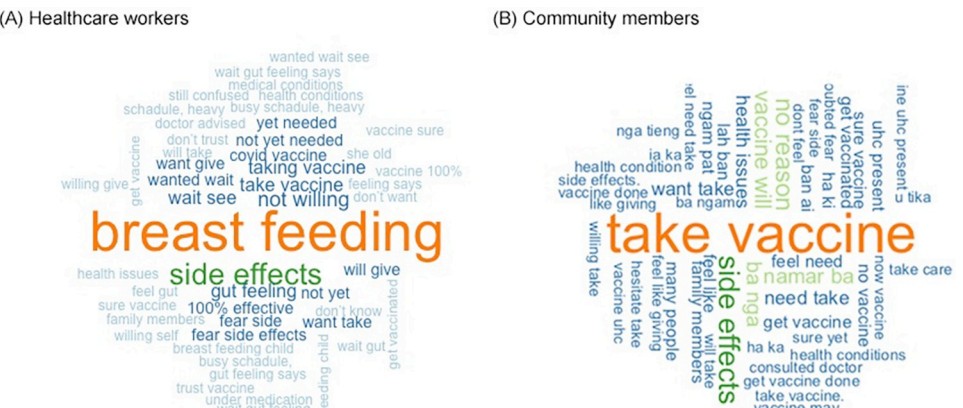

**Fig 1. Word cloud summary of reasons for COVID-19 vaccine hesitancy.**

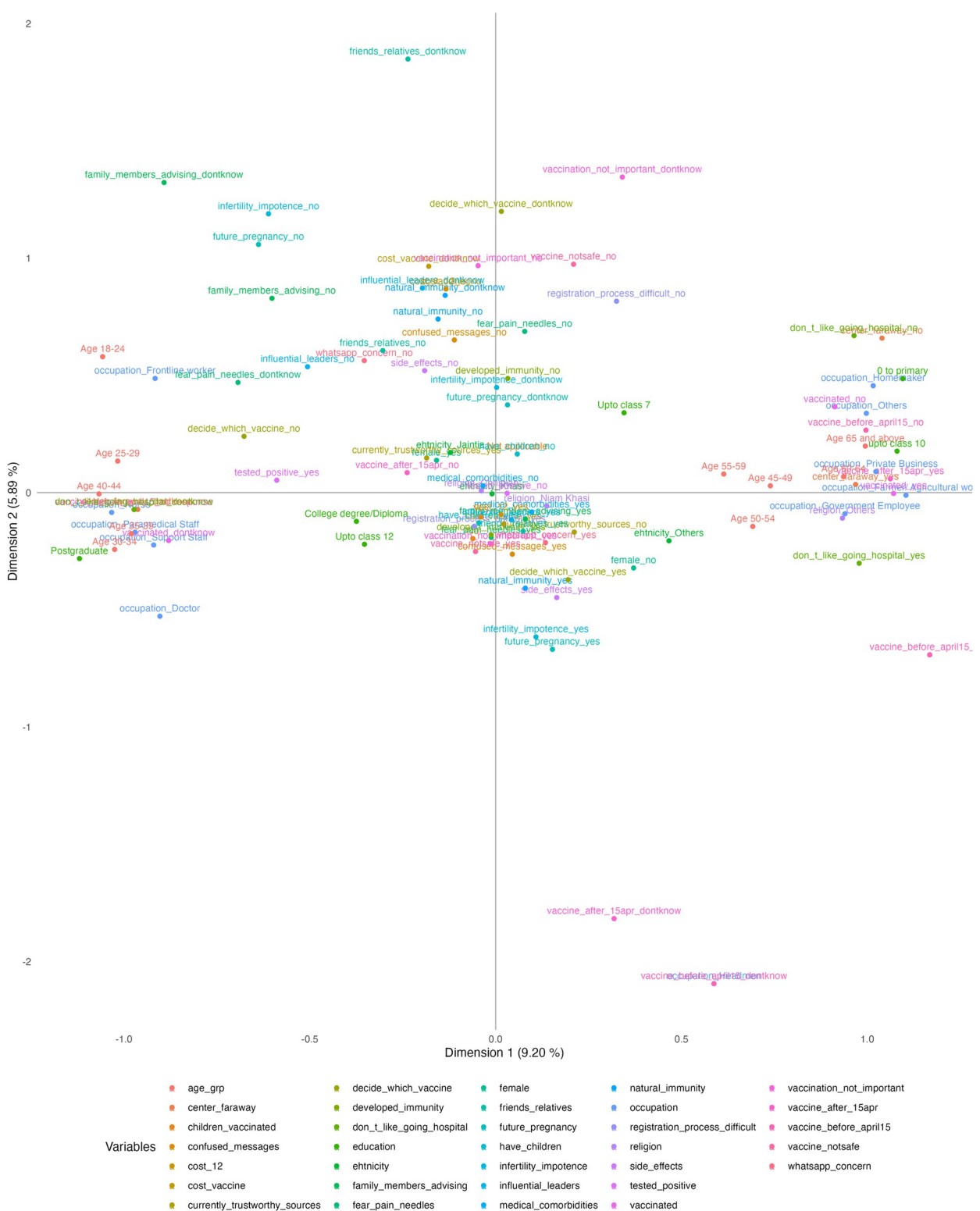

**Fig 2. Coordinates of all the variables on the first two dimensions of multiple correspondence analysis (MCA).**

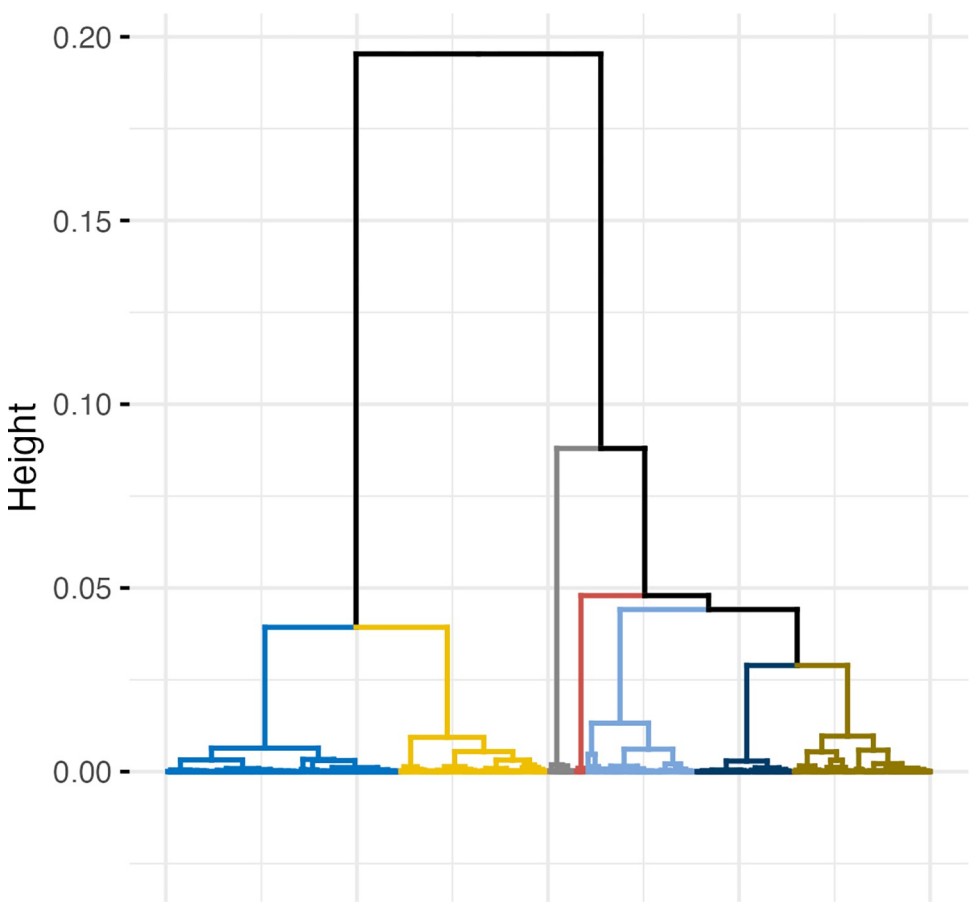

**Fig 3. Cluster dendrogram from agglomerative hierarchical clustering analysis (AHCA).**

Concerns about the cost of the vaccine, despite the fact that vaccines were being provided at no cost to the eligible population, and the complicated online registration process were commonly mentioned barriers to vaccination across all seven clusters. Long distances to vaccination centers were identified as another major barrier to vaccination among community members (clusters 3–7). Healthcare workers were divided into two distinct clusters, and both clusters had a similar distribution of healthcare workers by occupation. The first cluster of healthcare workers had a relatively lower prevalence of COVID-19-related misinformation compared to the other cluster. Their hesitancy seemed to be mainly driven by the difficulties with registration process, combined with opposition from family, peers, and influential leaders, and the concerns about vaccine safety. The second cluster of healthcare workers had a high prevalence of belief that COVID-19 is not a serious illness and that they have developed natural immunity. They were also concerned about vaccine safety and faced logistical challenges in obtaining appointments. Compared to the first cluster of healthcare workers, the second cluster had a notably higher percentage of individuals (>50%) who expressed concern over potential infertility and future pregnancy complications from vaccination. This is consistent with the demographic profile of the healthcare workers in this cluster where the majority were female and of reproductive age. Headmen of tribes or clans had a high prevalence of disbelief in COVID-19 and the majority reported not having any trustworthy source of information for COVID-19 vaccines. They were also more concerned about potential fertility-related

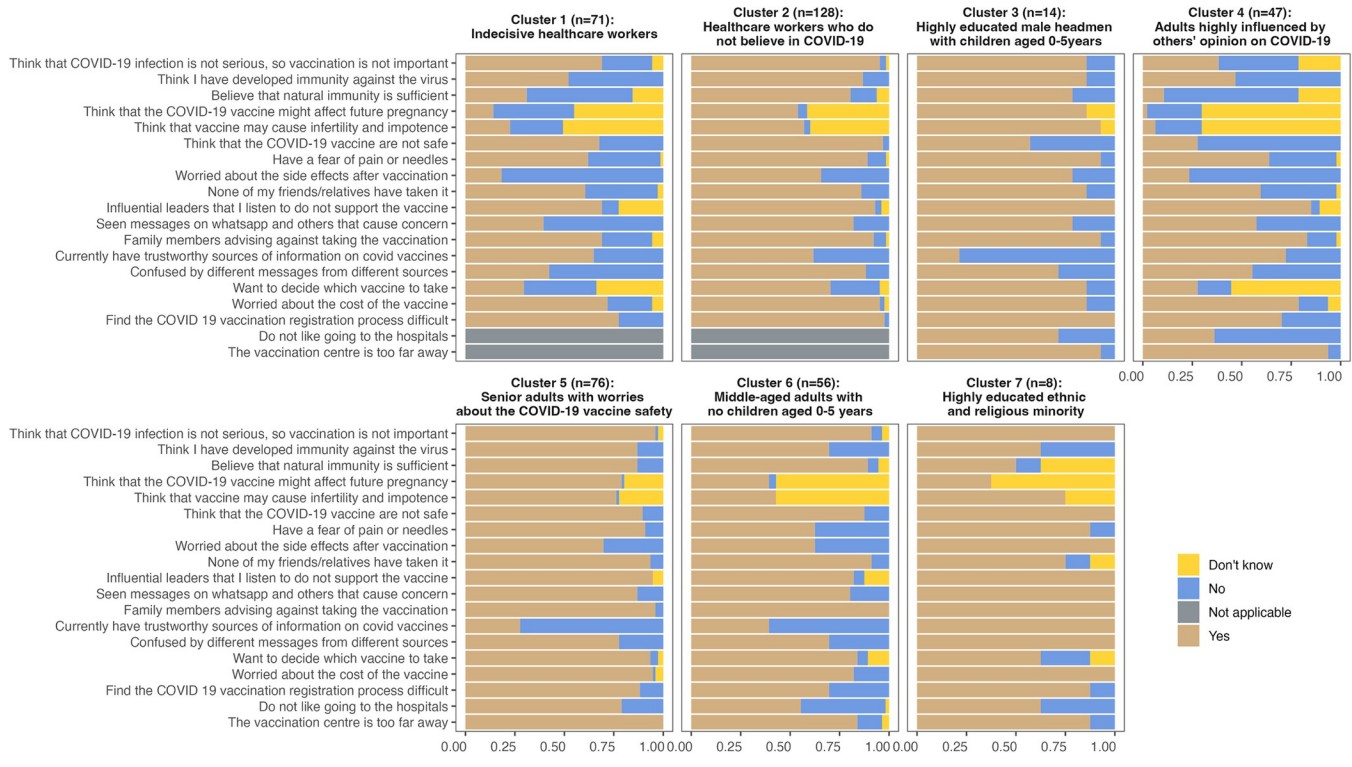

**Fig 4. Prevalence of negative perceptions of COVID-19 and vaccines by hesitant clusters.**

side effects of vaccines compared to the other clusters. The fourth cluster consisted of the general population with relatively low prevalence of negative perceptions towards COVID-19 or vaccines. However, similar to the first cluster of healthcare workers, the majority of participants in this group reported being advised against taking the vaccine by influential leaders and family members. The fifth cluster was composed of senior adults who were more concerned about vaccine safety than other groups. They also reported a high prevalence of all COVID-19-related negative perceptions, and reported a willingness to decide on their own which vaccine to take. The sixth cluster consisted of middle-aged adults without children aged 0–5 years, who lacked a trustworthy source for COVID-19-related information. They were advised by family members and influential leaders not to take the vaccine and reported a high willingness to decide on their own which vaccine to take. Over 95% of the individuals in this group, however, reported to have received the vaccine after April 15th, indicating low levels of hesitancy. The seventh and last cluster consisted of highly educated participants who identified themselves as ethnic and religious minorities. They reported being surrounded by high levels of misinformation from various sources, including influential leaders, WhatsApp messages, family members, and friends. This might have led them to believe that COVID-19 was not a serious illness, and COVID-19 vaccines were unsafe.

## Discussion

In this study, we used the multiple correspondence analysis in combination with an agglomerative hierarchical clustering analysis to identify distinct groups of vaccine-hesitant individuals among healthcare workers and community members residing in Meghalaya state, India. Within our sample of 400 survey participants, we observed seven distinct clusters. Despite all participants being

eligible to receive the COVID-19 vaccine free of charge, the perceived logistical challenges associated with receiving the vaccine were identified as a common barrier contributing to vaccine hesitancy across all seven clusters. These barriers included the long distances between vaccination centers and residences, the perceived cost burden of the vaccine, and the complicated registration process. Similarly, the influence of social networks, including family members, friends, and influential leaders, emerged as an important contributing factor to hesitancy across all the clusters. This is consistent with the findings from two previous studies conducted in Meghalaya, in which social and religious factors were identified as the main drivers of vaccine hesitancy [30,31]. Another study conducted in India also noted that individuals with populistic attitudes towards vaccines tend to be more hesitant towards and resistant to vaccination [41].

Second, the prevalence of COVID-19-related misinformation was high in five of the seven clusters, namely clusters 2, 3, 5, 6, and 7. In these clusters, over 90% of participants believed that COVID-19 was not a serious illness and therefore vaccination was not important. This is particularly noteworthy given that the survey took place at the beginning of the Delta-variant outbreak in the country in May 2021. One possible explanation is that the survey participants were not well-informed about the COVID-19 situation. Concurrently, these same clusters also exhibited very high levels of concern over vaccine safety. The observed trend of heightened vaccine hesitancy could potentially be attributed to the overwhelming prevalence of COVID-19-related misinformation in the country [41–43]. Additionally, the specific context in India before and during the survey might have exacerbated the safety concerns with COVID-19 vaccines, including the perceived lack of public transparency from regulatory bodies and the absence of phase 3 trial data of an indigenously-developed vaccine before its roll-out due to expedited approval processes [44,45].

Three clusters–namely, clusters 2, 3, and 5, stood out due to the individuals' concerns about potential adverse effects of vaccines on fertility and future pregnancy outcomes. Among them, cluster 2 had a high proportion of female healthcare workers of reproductive age, and cluster 3 was mainly characterized as a group of tribal headmen whose marital partners may be of reproductive age. This pattern is consistent with the established knowledge that fertility-related misinformation drives vaccine hesitancy among men and women of reproductive age [46]. Furthermore, female healthcare workers of childbearing age may find themselves at the intersection of heightened exposure to the adverse consequences of COVID-19 and personal concerns regarding vaccine safety and fertility, all of which compounded by societal norms and expectations regarding their professional duty to provide care and their roles within their families, creating a complex decision-making environment regarding vaccination [47,48]. Interestingly, cluster 5, which is comprised of primarily senior participants with more than half aged 60 years or older, also exhibited high levels of concern about vaccine safety and misinformation based on their responses to the 19 questions, potentially explaining the observed higher prevalence of misinformation on infertility and adverse pregnancy-related outcomes among this group. While more information is needed to understand this better, limited access to accurate information, lower educational attainment, generational differences in health beliefs, and potential cognitive factors associated with aging may be contributing factors to this age group's susceptibility to vaccine-related misinformation. Finally, cluster 7 consisted of a small group of individuals belonging to ethnic and religious minorities in Meghalaya state. Several studies conducted in the United States and the United Kingdom have indicated that ethnic and religious minority groups tend to report higher levels of vaccine hesitancy, often attributed to their lack of trust in local government authorities [49–51]. This is particularly important, as it highlights the need to consider such groups in local- and state-level policies and programs to achieve a more equitable distribution of COVID-19 vaccines and ensure that the health benefits of vaccines accrue to all segments of the society.

Our study findings provide valuable insights for local and state health authorities to effectively target distinct subgroups of vaccine-hesitant populations with tailored health messaging. However, it is important to note that our study has a number of limitations. First, we did not explore the prevalence of COVID-19 vaccine hesitancy in the study setting or across the identified clusters of vaccine-hesitant individuals. While the full potential of our study can be better realized if our findings are complemented by this additional information, the lack of data that is representative of the population in Meghalaya state and the scarcity of resources limited the scale of the survey. Second, the use of telephonic interviews, necessitated by pandemic-related restrictions, may have introduced limitations, such as sampling bias, response bias stemming from communication without visual context, and limited understanding of non-verbal cues. Nonetheless, we believe that these biases were minimized by the use of a structured questionnaire in our study. Lastly, our dichotomous survey format, comprising 19 questions probing participants' reasons for vaccine hesitancy, might have oversimplified the nuanced nature of vaccine hesitancy. The use of validated survey instruments with reliable scales designed to measure the multifaceted aspects of COVID-19 vaccine hesitancy could address this limitation and enhance the generalizability of our findings. Currently, there are validated scales that measure COVID-19 vaccine hesitancy [52,53], which would have allowed us to measure hesitancy in our target population and compare it with that of other studies. A qualitative analysis for an in-depth investigation of the emerging patterns in this study may significantly enrich our understanding of vaccine hesitancy in this population.

Prior to this study, only two other studies have examined the nature of COVID-19 vaccine hesitancy in Meghalaya state, both of which were limited in scope with a specific focus on tribal communities [30,31]. Being the first study to investigate COVID-19 vaccine hesitancy in healthcare and community settings, despite its limitations, this study represents a significant step toward understanding the reasons behind vaccine hesitancy among various subgroups in an area with low overall vaccination rates. Since our study was conducted in May 2021, it is worth noting that Meghalaya state continued to expand vaccine eligibility and coverage over the subsequent year, eventually providing at least one dose of the vaccine to all eligible populations. Nevertheless, achieving full vaccination with two doses for all remained a challenge, with various contributing factors, including vaccine hesitancy, logistical challenges, and reduced communication on the risks of COVID-19.

In this context, our findings remain relevant for local health authorities and highlight both common and distinct factors that shape vaccine hesitancy across different subgroups, calling for a tailored approach to counter it. Potential measures include technology-based health literacy interventions and mass media engagement for social mobilization along with broader health policy measures such as monetary incentives to encourage vaccination that may effectively address the distinct drivers of vaccine hesitancy in various subgroups [54]. Furthermore, our approaches and findings may inform future responses to disease outbreaks that require timely mass population vaccination campaigns, underscoring the importance of targeted and evidence-based community engagement strategies that consider the multifaceted nature of vaccine hesitancy.

## Supporting information

**S1 Checklist. PLOS inclusivity in global research questionnaire.**
(DOCX)

**S1 Text.** Table A. Vaccine hesitancy questionnaire Table B. Eigen value and percentage of variance explained by each dimension in the multiple correspondence analysis Fig A. Variable coordinates of multiple correspondence analysis (Dimensions 1 and 3) Fig B. Variable

coordinates of multiple correspondence analysis (Dimensions 1 and 4) Fig C. Variable coordinates of multiple correspondence analysis (Dimensions 1 and 5) Fig D. Variable coordinates of multiple correspondence analysis (Dimensions 2 and 3) Fig E. Variable coordinates of multiple correspondence analysis (Dimensions 2 and 4) Fig F. Variable coordinates of multiple correspondence analysis (Dimensions 2 and 5) Fig G. Variable coordinates of multiple correspondence analysis (Dimensions 3 and 4) Fig H. Variable coordinates of multiple correspondence analysis (Dimensions 3 and 5) Fig I. Variable coordinates of multiple correspondence analysis (Dimensions 4 and 5) Table C. Demographic and socioeconomic characteristics of survey respondents in each cluster
(DOCX)

**S1 Data.**
(CSV)

## Acknowledgments

We would like to express our gratitude to all the volunteers, Phiban Lyngdoh, Olivia Mazumder, Carmenia Khongwir, Peter J. Marbaniang, Carinthia B. Nengnong, Mattimi Passah, Quinnie Dorren Nongrum, Fellicita Pohsnem, Mary M. Rynjah, Alman Kshiar, Baiaineh Nongbri, Baiaineh Nongbri, Vicky Nelson Syiem, Fourness Lamin Amdep, Redolen Rose Dhar, Thangmaism Bikram Singha and Innangkyntiew Lyngdoh for their contribution to the data collection and entry process for this study.

## Author Contributions

**Conceptualization:** Sooyoung Kim, Rajiv Sarkar, Sampath Kumar, Sandra Albert.

**Data curation:** Rajiv Sarkar, Melissa Glenda Lewis, Sandra Albert.

**Formal analysis:** Sooyoung Kim.

**Investigation:** Sooyoung Kim, Rajiv Sarkar.

**Methodology:** Sooyoung Kim, Rajiv Sarkar, Melissa Glenda Lewis.

**Project administration:** Yesim Tozan, Sandra Albert.

**Supervision:** Yesim Tozan, Sandra Albert.

**Validation:** Sooyoung Kim, Rajiv Sarkar.

**Visualization:** Sooyoung Kim.

**Writing – original draft:** Sooyoung Kim.

**Writing – review & editing:** Sooyoung Kim, Rajiv Sarkar, Sampath Kumar, Yesim Tozan, Sandra Albert.

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
