## [Decision Letter · Decision Letter 0]

23 Oct 2023

PGPH-D-23-01274

Understanding COVID-19 Vaccine Hesitancy in Meghalaya, India: An Analysis of Cross-Sectional Survey Data Using Multiple Correspondence Analysis and Agglomerative Hierarchical Clustering

Dear Dr. Tozan,

Thank you for submitting your manuscript to PLOS Global Public Health. After careful consideration, we feel that it has merit but does not fully meet PLOS Global Public Health’s publication criteria as it currently stands. Therefore, we invite you to submit a revised version of the manuscript that addresses the points raised during the review process.

EDITOR'S COMMENTS:

Consider the points raised by the reviewers carefully and address them,A suggestion for title revision has been made, I believe that the study will benefit from such revision,Kindly address issues pertaining to how instrument developed for data collection was validated and reliability ensured. 

We look forward to receiving your revised manuscript.

Kind regards,

Nnodimele Onuigbo Atulomah, PhD

Academic Editor

Journal Requirements:

2. In the ethics statement in the Methods, you have specified that verbal consent was obtained. Please provide additional details regarding how this consent was documented and witnessed, and state whether this was approved by the IRB

3. Please include a complete copy of PLOS’ questionnaire on inclusivity in global research in your revised manuscript. Our policy for research in this area aims to improve transparency in the reporting of research performed outside of researchers’ own country or community. The policy applies to researchers who have travelled to a different country to conduct research, research with Indigenous populations or their lands, and research on cultural artefacts. The questionnaire can also be requested at the journal’s discretion for any other submissions, even if these conditions are not met.  Please find more information on the policy and a link to download a blank copy of the questionnaire here: https://journals.plos.org/globalpublichealth/s/best-practices-in-research-reporting. Please upload a completed version of your questionnaire as Supporting Information when you resubmit your manuscript.

4. Please send a completed 'Competing Interests' statement, including any COIs declared by your co-authors. If you have no competing interests to declare, please state "The authors have declared that no competing interests exist". Otherwise please declare all competing interests beginning with twhe statement "I have read the journal's policy and the authors of this manuscript have the following competing interests:"

5. Please provide separate figure files in .tif or .eps format only and remove any figures embedded in your manuscript file. Please also ensure all files are under our size limit of 10MB.

6. We have noticed that you have uploaded Supporting Information files, but you have not included a list of legends. Please add a full list of legends for your Supporting Information files after the references list.

7. In the online submission form, you indicated that "The data that support the findings of this study are available on request from the corresponding author, SA". All PLOS journals now require all data underlying the findings described in their manuscript to be freely available to other researchers, either 1. In a public repository, 2. Within the manuscript itself, or 3. Uploaded as supplementary information.

Additional Editor Comments (if provided):

The manuscript has been reviewed and considered as an important contribution be included in the field to provide some understanding of the emerging challenges associated with Vaccine Hesitancy. However, to make the manuscript acceptable for publication, certain revisions have been suggested. Kindly pay attention to these suggestions as these would strengthen the manuscript more and clarify issues appropriately.

Title: The current title appears too long and needs modification. Suggested modification reads thus: "Elucidating the dynamics involved in COVID-19 Vaccine Hesitancy in Meghalaya, India: An Agglomerative Hierarchical Cluster Analysis".

Introduction: The last two sentences of the introduction mention "...The issue of vaccine hesitancy has become more apparent during a global pandemic and in the face of a new vaccine to address it" What science explains the dynamics involved in vaccine hesitancy? This has not featured in the study theoretical principles. It is expected that vaccine hesitancy is a personal-level disposition that triggers refusal or acceptance. The theoretical framework elucidating the disposition is completely omitted and not considered for evaluation in the study.

Methods: The instrument development process appears missing including methods applied for assuring validity and reliability. The variables measured are not explicitly indicated in the methodology, but somewhere in figure 4 it gives the impression that negative perceptions of COVID-19 were measured. How was perception conceived and operationalized in the methodology of instrument development?

Reviewers' comments:

Reviewer's Responses to Questions

**Comments to the Author**

1. Does this manuscript meet PLOS Global Public Health’s publication criteria? Is the manuscript technically sound, and do the data support the conclusions? The manuscript must describe methodologically and ethically rigorous research with conclusions that are appropriately drawn based on the data presented.

Reviewer #1: Yes

Reviewer #2: Yes

2. Has the statistical analysis been performed appropriately and rigorously?

Reviewer #1: Yes

Reviewer #2: Yes

3. Have the authors made all data underlying the findings in their manuscript fully available (please refer to the Data Availability Statement at the start of the manuscript PDF file)?

Reviewer #1: Yes

Reviewer #2: No

4. Is the manuscript presented in an intelligible fashion and written in standard English?

Reviewer #1: Yes

Reviewer #2: Yes

5. Review Comments to the Author

Reviewer #1: The manuscripts highlights important factors in the fight against COVID-19, particularly vaccine hesitancy and associated factors. This research is a thorough research that was done and the results well presented. The findings has potentials to impact public health policy and educational as well as health communication program design and implementation.

The objective of the research and manuscript is very clear. A robust statistical analysis procedures were used in analyzing and presenting the findings. Again, the authors have presented a discussion section that relates well with existing limited literature in the COVID-19 research and will had positive impact on knowledge and practices.

I do recommend that the language of the manuscript especially the methods section be examined again and effort needs to be made tp present the methods section with lay person's language and reading level. It appears to be verbose and use of technical terminology are used. This will make it a challenging for the general academic public to understand. Secondly, figures 2 and 4 don't appear clear and readers may not appreciate the importance of the messages in the figures. Please, make the figures more clear and simple to understand.

Overall, the author have presented a very elaborate research design and findings that only informs the Indian authorities and public about the determinants of covid-19 vaccine hesitancy, but the whole world, especially in developing countries where more misinformation, disinformation, and cultural barriers are reported to be challenges of health behavior.

Reviewer #2: Overall, the article is well-structured and informative. However, there are a few areas where improvements could be made for publication.

Title of the study

• It's a bit long and detailed, which might make it less reader-friendly. Consider simplifying it while retaining the essential information.

Abstract

• Ensure that the abstract is concise while retaining clarity. Some sentences, such as those introducing the identified clusters, could be streamlined for brevity without sacrificing essential information.

• Specify the timeframe of the survey administration. Knowing when the survey was conducted is important, especially when dealing with evolving situations like the COVID-19 pandemic.

Background of the study

• The mention of disparities in the quality of COVID-19 data reporting is crucial, but the article could elaborate on how this may impact the understanding of vaccine hesitancy.

• While the authors acknowledge regional disparities in COVID-19 infection and vaccination rates, they could delve deeper into the specific factors contributing to these disparities, beyond stating that they are highly clustered in more urbanized and prosperous states.

• While the study provides valuable insights into vaccine hesitancy, incorporating a theoretical framework would have enhanced the robustness and interpretability of the findings by offering a structured lens through which to analyze and understand the complex factors influencing vaccine hesitancy in diverse subpopulations.

Method

• The study period is mentioned as May 2021, but the recruitment of participants is stated as May 1st, 2022, to May 22th, 2022. There seems to be a discrepancy; please clarify the correct dates for the study period and recruitment.

• It would be helpful to provide information on the sample size determination process and power analysis to ensure that the study has sufficient statistical power to detect meaningful differences within subgroups.

• While telephonic interviews were necessary due to pandemic-related restrictions, acknowledge and discuss potential limitations introduced by this mode of data collection.

• The dichotomous (yes or no) response format for the 19 questions might oversimplify the complexity of vaccine hesitancy.

• While verbal consent is mentioned, it would be beneficial to elaborate on the specifics of the consent process, ensuring participants fully understand the purpose of the study, their rights, and the potential implications of their participation.

• While you've described the methods used in data analysis (e.g., agglomerative hierarchical clustering based on Ward’s method), providing a brief rationale or justification for choosing these methods would add depth to the paper. This could help readers understand why these specific analytical approaches were employed.

Results

• It might be helpful to use subheadings to distinguish between different sections of the results, making it easier for readers to follow the flow of information.

• For each cluster, provide more context on why certain characteristics or attitudes are prevalent within that group. This could involve a deeper exploration of the socio-cultural context in Meghalaya that may contribute to these patterns of vaccine hesitancy.

• Use consistent terminology throughout the paper. For example, ensure that terms like "undecisive" are used consistently, and consider alternatives such as "indecisive" for clarity.

Discussion of Findings and Addressing Study Limitations

• Explore further the gender and age dynamics within clusters, especially Cluster 2 with female healthcare workers and Cluster 5 with senior participants. Provide insights into how age and gender intersect with vaccine hesitancy.

• Given that the study was conducted in May 2021, consider discussing the relevance and timeliness of the findings in the context of the evolving COVID-19 situation. This is particularly important if there have been significant changes in vaccine distribution, public perception, or policies since the study period.

• Acknowledge the study's limitations, such as the lack of data representative of the entire population in Meghalaya and the use of non-validated survey instruments. Discuss how these limitations might affect the generalizability of the findings.

Proposing Future Research Directions:

• Suggest potential avenues for future research based on the gaps identified in this study.

6. PLOS authors have the option to publish the peer review history of their article (what does this mean?). If published, this will include your full peer review and any attached files.

**Do you want your identity to be public for this peer review?** For information about this choice, including consent withdrawal, please see our Privacy Policy.

Reviewer #1: No

Reviewer #2: No

---

## [Editor Report · Decision Letter 1]

29 Jan 2024

Understanding COVID-19 Vaccine Hesitancy in Meghalaya, India: Multiple Correspondence and Agglomerative Hierarchical Cluster Analyses

PGPH-D-23-01274R1

Dear Dr. Tozan,

We are pleased to inform you that your manuscript 'Understanding COVID-19 Vaccine Hesitancy in Meghalaya, India: Multiple Correspondence and Agglomerative Hierarchical Cluster Analyses' has been provisionally accepted for publication in PLOS Global Public Health.

Best regards,

Nnodimele Onuigbo Atulomah, PhD

Academic Editor

The reviewers' recommendations have all been performed successfully to the satisfaction of the academic editor.